# Determination of Organic Compounds, Fulvic Acid, Humic Acid, and Humin in Peat and Sapropel Alkaline Extracts

**DOI:** 10.3390/molecules26102995

**Published:** 2021-05-18

**Authors:** Laurynas Jarukas, Liudas Ivanauskas, Giedre Kasparaviciene, Juste Baranauskaite, Mindaugas Marksa, Jurga Bernatoniene

**Affiliations:** 1Department of Analytical and toxicological chemistry, Lithuanian University of Health Science, Kaunas, Sukileliu av. 13, LT-50161 Kaunas, Lithuania; laurynas.jarukas@lsmuni.lt (L.J.); liudas.ivanauskas@lsmuni.lt (L.I.); juste.baranauskaite@lsmuni.lt (J.B.); mindaugas.marksa@lsmuni.lt (M.M.); 2Department of Drug Technology and Social Pharmacy, Lithuanian University of Health Sciences, LT-50161 Kaunas, Lithuania; jurga.bernatoniene@lsmuni.lt

**Keywords:** organic compounds, fulvic acid, humic acid and humin, peat

## Abstract

Black, brown, and light peat and sapropel were analyzed as natural sources of organic and humic substances. These specific substances are applicable in industry, agriculture, the environment, and biomedicine with well-known and novel approaches. Analysis of the organic compounds fulvic acid, humic acid, and humin in different peat and sapropel extracts from Lithuania was performed in this study. The dominant organic compound was bis(tert-butyldimethylsilyl) carbonate, which varied from 6.90% to 25.68% in peat extracts. The highest mass fraction of malonic acid amide was in the sapropel extract; it varied from 12.44% to 26.84%. Significant amounts of acetohydroxamic, lactic, and glycolic acid derivatives were identified in peat and sapropel extracts. Comparing the two extraction methods, it was concluded that active maceration was more efficient than ultrasound extraction in yielding higher amounts of organic compounds. The highest amounts of fulvic acid (1%) and humic acid and humin (15.3%) were determined in pure brown peat samples. This research on humic substances is useful to characterize the peat of different origins, to develop possible aspects of standardization, and to describe potential of the chemical constituents.

## 1. Introduction

Humic substances are complexes of bioactive substances of microbiological, vegetative, or animal origin that are widely spread in nature. The rich sources of these compounds are soil, humus, peat, sapropel, natural water, and various other environments [1]. Humic substances are organic macromolecules with multiple properties and high structural complexity. Usually, they are divided into three components based on their solubility: fulvic acids, humic acids (alkali-soluble), and humin (insoluble residue). They contain major functional groups, including carboxylic, phenolic, carbonyl, hydroxyl, amine, amide, and aliphatic moieties, among others [1,2,3,4]. Due to the specificity of chemical properties, humic substances are applicable in industry, agriculture, the environment, and biomedicine with well-known and novel approaches.

Humic acids have been used for centuries in the Indian Ayurvedic tradition and in other countries as a complementary and alternative medical choice for the promotion of health in inflammatory conditions or metabolic disorders. Humus and its components are regarded as something pure, organic, and essential for the detoxification and functioning of the human body. Humic materials in aquatic systems and water sediments have been observed to be closely connected with the efficacy of hydrotherapy and balneotherapy [1,5]. Peat mud extracts used in balneotherapy were investigated for the specific activity of radioisotopes as a possible biological mechanism of action [6]. Many recent medical publications have stated that humic substances have biological activities, to protect against cancer and related cancer-causing viruses; can act as antiviral and anti-inflammatory agents; act as an active ingredient in wound healing; and exhibit antimutagenic/desmutagenic potential [1,7,8,9]. Specific studies showed reversal of cancers and tumors using special humic substance therapies [10,11,12]. Moreover, humic acid’s molecules, which assume a negative charge in neutral to basic media, can inhibit virus replication by binding cationic domains of the virus, which are necessary for virus attachment to the cell surface [13]. The humic acids are indeed global fertilizers of microbial growth, as proposed by the traditional view, and lead to an increase of more than 30% in the mean concentrations of the colonic microbiome [14]. The potential of humic substances to form chelate complexes with heavy metals (such as cadmium) enables them to be used for the elimination of heavy metals from living organisms [1], with continuous exploration in medicine studies [15]. The phenolic groups in humic substances act as electron-donating agents, scavenging free radicals and preventing chain reaction initiation. Indirectly, humic acids could act as solubilizing agents, carrying pharmaceutical and cosmetic active ingredients in their micelle-like structures to enhance their water solubility [7]. These antioxidant and solubilizing properties are useful in cosmetic and pharmaceutical applications.

The humic substance system is formed by the association of various components present in the humification process, such as amino acids, lignins, pectins, or carbohydrates, through intermolecular forces (donor–acceptor, ionic, hydrophilic, and hydrophobic) [1]. The chemical structure is very complicated and depends on geographical, climatic, physical, and biological circumstances, respectively. The natural sources of humic substances in this research were peat and sapropel from Lithuania. Peat deposits found in many places around the world can be a significant recourse for humic substances. Peat is organic soil formed as a result of the incomplete disintegration and humification of marsh plants under high humidity conditions and can contain up to 40% of humic substances [16]. Peat is exposed to a relatively high oxygen level, which leads to the intensification of oxidation processes and to changes in the chemical and biological characteristics of humic substances [17]. Sapropel is a mixture of organic and inorganic materials washed into lakes from catchments and generated within the lakes. It contains all macroelements and microelements necessary for plants, as well as biologically active substances, such as vitamins, enzymes, and antibiotics [18,19,20].

Research carried out in various countries suggests that the efficacy of humic acid depends on its extremely complex chemical structure, which makes biochemical investigations elaborate, costly, and difficult to reproduce [7]. Many fundamental questions relating, in particular, to the physicochemical characteristics of humic and fulvic molecules are yet to be answered [21,22]. Differences in the obtained values have been attributed to either the variability of humic substances or the intrinsic limitations of methods when applied to poly-disperse humic systems [23,24]. Natural environmental sources for humic acids need characterization and standardization of their physical and chemical parameters. The goal of this study was to identify the main organic compounds and determine fulvic acid, humic acid, and humin concentration in different peat and sapropel extracts from Lithuania. The structure of humic and fulvic acids is dynamic regarding the origin, different experimental conditions, and methods; therefore, the investigation of these compounds is of great importance for pharmaceutical and biomedical applications.

## 2. Results and Discussion

### 2.1. Determination of Organic Compounds

Peat and sapropel samples were extracted using different techniques and a gas chromatography–mass spectrometry (GC-MS) analysis was performed. Chromatograms of extracts were obtained through the conditions described, and determination of various organic compounds was achieved. The highest amounts of organic compounds were determined in the light peat extract after active extraction; therefore, this chromatogram is presented in this paper (Figure 1a). Detailed chromatogram data are provided in Figure 1b. A total of 63 organic compounds were identified in light peat, and the main organic acid was lactic acid derivative—22.46% of the total distribution amount. Other organic acid derivatives were glycolic, hexacosanoic, lignoceric, and glyceric acids, which accounted for more than 1% of the total amount. The rest of the components were salts, esters, and other organic derivatives. The main compound in the brown peat extract was bis(tert-butyldimethylsilyl) carbonate—22.30% of the total distribution amount, besides significant amounts of lactic and glycolic acid derivatives. The highest amount of bis(tert-butyldimethylsilyl) carbonate (25.68%) was determined in black peat extract. It should be noted that bis(tert-butyldimethylsilyl) carbonate was not detected in the extracts of sapropel, while the dominant compound was malonic acid amide. The variety of organic compounds is large: it is known that their major functional groups include carboxylic, phenolic, carbonyl, hydroxyl, amine, amide, and aliphatic moieties, among others [1].

A common list of the main organic compounds determined in different peat and sapropel extracts is shown in Table 1. Other compounds were in very low concentrations or just traces detected and are therefore not shown in the table. The highest mass fraction of the compounds was bis(tert-butyldimethylsilyl) carbonate; it was in all the tested peat extracts in variations from 6.90% to 25.68% but was not detected in sapropel extracts. The amount of lactic and glycolic acid derivatives was lowest in black peat, higher in brown peat, and highest in light peat extracts. Acetohydroxamic acid was determined in all peat and sapropel extracts, but the highest amount, 8.10%, was in sapropel after active maceration (*p* < 0.05). Oxalic acid derivative and glyceric acid were determined in almost all extracts and varied very similarly in small quantities. It was interesting to note that lignoceric and hexacosanoic acid derivatives were found only in the peat samples after active maceration, while they were not detected in samples after ultrasound extraction.

Comparing two extraction methods, it was determined that active maceration was more efficient in yielding higher amounts of organic compounds. Total organic compound distribution using AME was higher than that using USE—in black peat extracts, 2.6 times; in brown peat extracts, 2.4 times; in light peat extracts, 1.4 times; and in sapropel extracts, 2 times (*p* < 0.05).

GC-MS analysis determined from 28 to 53 constituents (amount more than 1%) in the black peat extracts, from 47 to 49 constituents in the brown peat extracts, from 57 till to constituents in the light peat extracts, and from 41 to 52 constituents in the sapropel extracts (data not shown). Comparison of determined organic compounds by the GC-MC method is very limited due to the specific substance and variety of methods used in similar research. GC-MC analysis of brown coal samples (China) determined 47 organic compounds, but just a few are the same as this analysis: acetic acid and butenedioic acid derivative [8].

The composition of peat is a mixture of the decomposition products of plant and animal residues and of substances synthesized biologically and/or chemically from decomposed products or intermediate products [25]. Therefore, chemical characterization may vary greatly and depend on different factors.

Published studies concerning the characterization of humic substances demonstrate various research studies for the chemical structure or physical parameters in many ways and methods. Physical characterization of peat samples (Russia) was analyzed with the infrared (IR) absorption spectra for the functional groups and fluorescence measurements for the humification index. The average molecular weights of the HA fractions were determined by high-performance size-exclusion chromatography (HP-SEC) [16]. The elemental composition of humic substances was detailed and showed that the major elements are carbon, hydrogen, oxygen, nitrogen, and sulfur by X-ray fluorescence, and another way—by combustion on a Carlo Erba Strumentazione Model 1106 C, H, N analyzer [16,26,27,28]. Chemical group composition can also be used to characterize humic substances. Natural organic matter of soil consists of polyphenolic, carbohydrates, and other low-molecular-weight organic compounds determined by various spectrophotometric methods [29]. Many researchers agree that the chemical structure of humic substances is very complicated and depends on their source and origin [1,7,16,29].

### 2.2. Determination of Fulvic Acid, Humic Acid and Humin

Results of the fulvic acid amount expressed as a percentage of the total compound distribution are shown in Figure 2. The highest amount—1%—was determined in pure brown peat samples; a similar amount in black and light peat; and the lowest in sapropel (*p* < 0.05). Peat extracts after ultrasound-assisted extraction contained the highest amount of fulvic acid (0.53%) prepared from light peat; and the lowest, from sapropel. The highest result after active maceration was determined in the brown peat extract (0.45%), and the lowest in black peat and sapropel extracts. Qin et al. determined a similar amount to this research of fulvic acid—1.07% in brown coal of China origin [8]. Gondar published Spanish peat characterization with a low quantity: fulvic acid determination in the 0.014–0.07% range [30].

Results of humic acid and humin amounts expressed as a percentage of total compound distribution are shown in Figure 3. The highest amount—15.3%—was determined in the pure brown peat samples; this result coincided with that of the fulvic acid amount (Figure 3). The lowest amount of humic acid and humin was determined in pure black peat and sapropel. Similar to our results, the amount of humic acids was determined in Spanish peat and varied from 1.1% to 8% [30]. Comparing the two extraction methods, a statistically higher amount was extracted by active maceration in all tested samples. The highest quantity of substances was in brown peat extract, and the lowest in sapropel, using both extraction methods (*p* < 0.05).

The comparison of extract analysis showed higher amounts of humic acid and humin than of fulvic acid. Studies on particle size measurement of humic substances determined that humic particles were smaller that fulvic acid; the bigger diameters of submicron particles observed for fulvic acids in comparison with humic ones could be caused by higher amounts of ionizable functional groups, allowing for the greater expansion of fulvic coils in alkaline solution [31]. Such physical characteristics can influence higher extraction of humic acid, as shown by our results. The results of humic substance evaluation showed a positive correlation between the amount of fulvic acid and the amount of humic acid and humin *r* = 0.7541 (*p* = 0.0046). Other researchers maintain that the ratio of fulvic to humic acid varies between soils and between horizons of the same soil. Humic fractions are involved with solubilization of the sesquioxide; therefore, additional steps of separation are recommended [25]. Results for humic substances determined in the pure peat, sapropel, and alkaline extracts were not consistent. Fulvic acid amount was higher in pure black and brown peat, and humic acid and humin amount was higher in pure brown and light peat in comparison with alkaline extracts. These variations demonstrate that the extraction procedures should be modified and optimized to achieve an expanded analysis of humic substances.

## 3. Materials and Methods

### 3.1. Materials

Samples of peat and sapropel were obtained from the joint stock company “Durpeta” (one of the oldest and still operating peat extraction and processing companies in Lithuania). The sources of the samples were peat bogs situated in areas of southeastern Lithuania. Three peat samples were analyzed: light, brown, and black. Color is produced by the minerals present and by the organic matter content, or it is characteristic of organic materials according to their degree of decomposition. Dark brown or black color of peat indicates that it has high organic matter content and is the most highly decomposed. The materials with intermediate degree of decomposition are brown, and those with the lowest degree of decomposition are commonly light yellowish-brown.

Analytical quality reagents (Merk Co., Sigma-Aldrich Co., Fluka Chemie AG RdH Laborchemikalien GmbH Co., Seelze, Germany) were used without purification. Solutions were prepared from high-purity water (Millipore Elix 3, Millipore Co. Molsheim, France), produced at 10–15 MΩ cm.

### 3.2. Preparation of Extracts

A sieved fraction of peat or sapropel particles of 2–5 mm size was used for the extraction. Extraction was performed with 1% NaOH solution. A material and solvent ratio of 2:30 was applied. Extraction was performed using a conical flask covered with parafilm for 2 h. Ultrasound extraction (USE) was performed using ultrasonic activation of 0.200 kW (ultrasonic bath Digital XUB-10) at 40 ± 2 °C. Active maceration (AME) was performed using heating and stirring with the magnetic stirrer, maintaining the boiling point at 92–95 °C. At the end of the extraction, the flask was cooled to 25 ± 2 °C. The extracted mixtures were centrifuged for 4 min (centrifuge SIGMA 3-18KS, Seelze, Germany) at 3000 rpm after decanting and filtering. The final volume of 50 mL was adjusted with 1% NaOH solution.

### 3.3. Organic Compound Determination by GC-MS

Derivatization and GC-MS analysis were performed according to the previous research about the analysis of organic compounds from natural sources [32].

Sample derivatization procedure: In total, 0.1 g of the prepared extract solution was evaporated to dryness with a stream of nitrogen gas. Briefly, in a 2 mL ampoule bottle, 0.1 g of dried extract sample was diluted into 0.1 mL of extraction solvent (acetonitrile), and 0.1 mL of the derivatization agent N-tert-Butyldimethylsilyl-N-methyltrifluoroacetamide (MTBSTFA) was added in sequence. The vial was sealed and oscillated by a vortex mixer for 1 min, and then, to allow the mixture to react, it was placed in a glycerol bath at 130 °C for 90 min. The subsequent solution was transferred to 200 μL autosampler vials with inserts, and 2 μL aliquot was injected into the GC-MS system for analysis. Efficiency extraction parameters were evaluated and optimized, including derivatization time, extraction temperature, and reagent amount on derivatization.

GC-MS method: Analyses were performed using a SHIMADZU GC/MS-QP2010nc (Shimadzu, Japan). A robotic autosampler and a split/splitless injection port were used. The injection port temperature was kept at 250 °C until the end of analysis. The separation of analytes was carried out on an Rxi-5 MS (Restek Corporation capillary column (30 m long, 0.25 mm outer diameter, and 0.25 μm liquid stationary phase thickness) with a liquid stationary phase (5% diphenyl and 95% polydimethylsiloxane)) with helium at a purity of 99.999% as the carrier gas in a constant flow of 1.49 mL/min. The oven temperature was programmed at 75 °C for 5 min, then increased to 290 °C at 10 °C/min and increased to 320 °C at 20 °C/min and kept for 10 min. Total time was 41 min. The temperatures of the MS interface and the ion source were set at 280 and 200 °C, respectively. The MS was operated in the positive mode (electron energy 70 eV). Full-scan acquisition was performed with the mass detection range set at 35–500 *m*/*z* to determine the retention times of analytes, to optimize oven temperature gradient, and to observe characteristic mass fragments for each compound. For the identification and quantification of analytes, the total ion current (TIC) mode was used. Data acquisition and analysis were executed using LabSolution GC/MS (version 5.71) (Shimadzu Corporation). The components were identified using the National Institute of Standards and Technology (NIST) mass spectral library (NIST14, NIST14s) and the Wiley Registry of Mass Spectral Database (WR10, WR10R), with a similarity of ≥ 95%. Amounts of compounds are presented as a percentage of the peak area in the tested sample.

### 3.4. Determination of Fulvic Acid, Humic Acid, and Humin

Humic substances were analyzed according to the recommendations of agricultural chemical analysis [25].

Procedure. In total, 10 g of air-dry sieved (5 mm) peat was weighed into a 250 mL plastic centrifuge bottle, and 200 mL 0.5 M NaOH was added and shaken overnight. The centrifugation process (at 2000 rpm for 20 min) was used to allow sedimentation of the insoluble humin and all of the supernatant was transferred into a clean centrifuge bottle. Into the supernatant solution, 6 M HCl was added to adjust to pH 2.0. The centrifugation process was used to sediment the humic acid. The decant solution mainly composed of fulvic acid was placed into a 250 mL volumetric flask and used for quantitative analysis.

UV–VIS spectrophotometry for fulvic acid determination: The optical density of the prepared solution was read at 465 nm wavelength (if necessary, dilution should be done to bring it on scale) with the UV–VIS spectrophotometer HALO DB-20 (Dynamica, Livingston UK). The approximate concentration of fulvic acid in mg/100 mL is given by comparing with the graph of optical density vs. concentration. Original graph of the relationship between optical density at 465 nm and concentration of ash-free sedge peat fulvic acid is presented in the used reference.

Calculation. The amount of ash-free fulvic acid in mg/100 mL is read from the chart. This solution resulted from 10 g air-dry soil in 250 mL solution; therefore, 250 mL solution contains y × 250/100 mg fulvic acid, which converts to *25 y* mg fulvic acid 100 g^−1^, or *0.025 y*% air-dry soil. This must be multiplied by any dilution factor before reading the optical density.

Continuing procedure: For humic acid determination, 30 mL of 0.5 M HCl was added into the sediment part of the sample, and the centrifugation process was used. The supernatant was transferred into the volumetric flask and washed with distilled water. The same procedure was repeated two times. Then, 60% ethanol was added into the washed sample and transferred into a pre-weighed oven-dry 100 mL glass beaker. The solution was evaporated to dryness carefully on a hotplate, avoiding loss by spitting, and cooled in a desiccator and reweighed. The difference in weights gives the weight of humic acid plus ash. The sample was ignited in a muffle furnace at 500 °C overnight to burn off the humic acid fraction and cooled in a desiccator, and then the beaker containing the residual ash was reweighed. This weight was subtracted from the weight of the beaker and residue before ashing to obtain the weight of ash-free humic acid. The weight procedure was performed with the semi-microanalytical balance AUW120D (Shimadzu, Tokyo, Japan).

Calculation. The amounts of humic acids and humin are calculated using gravimetric determination results, using the formula:$$HA\%=\left[\frac{Weight\;of\;ash-free}{Test\;proportion\;dry\;weight}\right]\times 100$$

### 3.5. Statistical Analysis

The results are presented as mean ± standard deviation. Statistical analysis was performed by a paired *t*-test and Pearson correlation using the software package Prism (GraphPad Prism 8 Software Inc., La Jolla, CA, USA). A value of *p* < 0.05 was taken as the level of significance.

## 4. Conclusions

Different peat and sapropel extracts of Lithuania were analyzed for the potential source of humic substances. Organic compound composition determined by the GC-MS method presented high amounts of bis(tert-butyldimethylsilyl) carbonate and lactic acid derivatives in peat extracts, and malonic acid derivatives in sapropel extracts. The quantity of fulvic acid was determined spectrophotometrically; the highest amount was determined in the brown peat sample. Humic acid and humin were analyzed by thermo-gravimetric studies, and the highest amounts were in brown peat samples. Peat and sapropel samples were extracted with 1% NaOH solution by two different methods, and the results showed that active maceration was more effective than ultrasound-assisted extraction. The present study confirms the wide composition of organic substances and humic substance variations in different sources and the influence of extraction conditions. Therefore, future research of natural organic matter will aim at specifying a dominant compound or group as markers for the standardization and development of bioactive products.

## Figures and Tables

**Figure 1 molecules-26-02995-f001:**
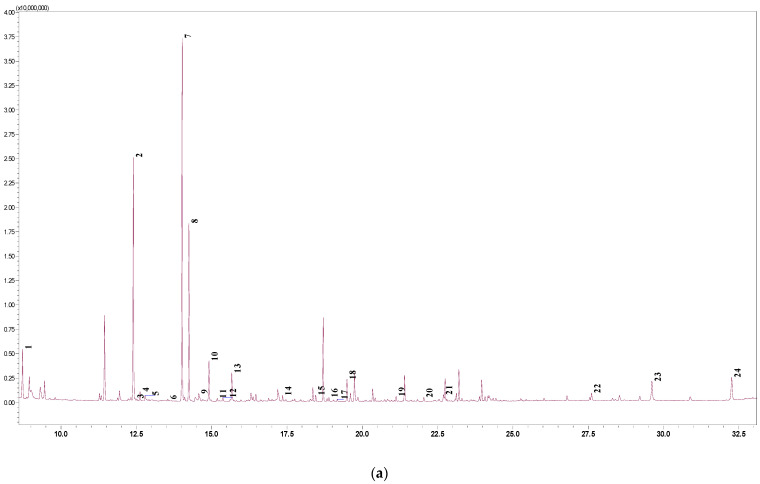
(**a**) GC-MS analysis chromatogram of the light peat alkaline extract after active maceration. (**b**) List of determined compounds of the GC-MS analysis chromatogram of the light peat alkaline extract after active maceration.

**Figure 2 molecules-26-02995-f002:**
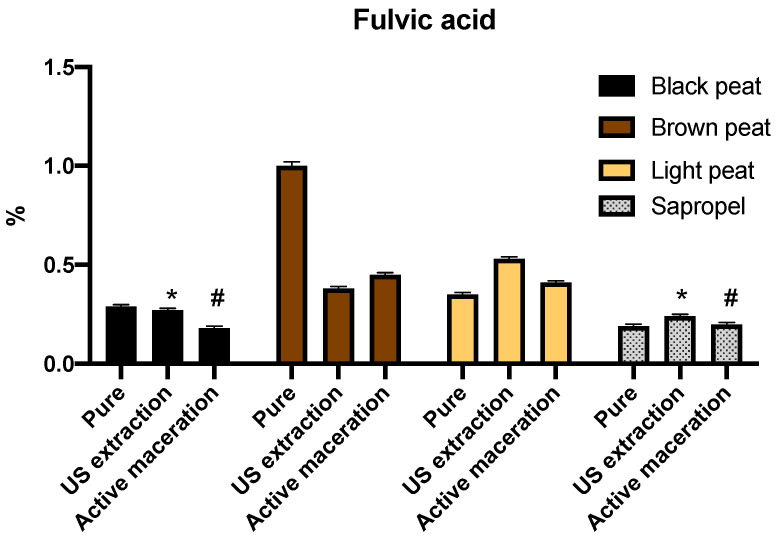
Fulvic acid amount (percentage of total compound distribution) determined in the pure peat, in the extracts after ultrasound-assisted extraction and active maceration. *—Values with *p* = 0.1217; #—values with *p* = 0.2254; all other values with *p* < 0.05.

**Figure 3 molecules-26-02995-f003:**
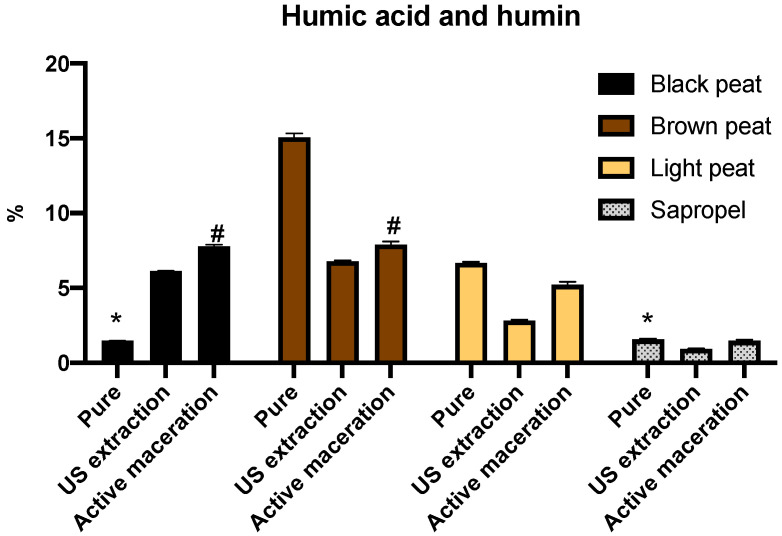
Humic acid and humin amount (percentage of total compound distribution) determined in the pure peat, in the extracts after ultrasound-assisted extraction and active maceration. *—Values with *p* = 0.1472; #—values with *p* = 0.5203; all other values with *p* < 0.05.

**Table 1 molecules-26-02995-t001:** Organic compound percentage distribution (%) in different peat and sapropel extracts.

Organic Compound Percentage Distribution (%)								
	Black Peat		Brown Peat		Light Peat		Sapropel	
	USE *	AME **	USE	AME	USE	AME	USE	AME
Cyclopentasiloxane, decamethyl-	2.61	4.15	3.32	4.93	5.23	3.25	1.37	1.15
Bis(tert-butyldimethylsilyl) carbonate	6.90	25.68	11.63	22.30	9.64	14.99	-	-
Malonic acid amide Aq	2.25	0.84	2.42	1.23	13.76	0.58	12.44	26.84
Lactic acid derivative	0.51	3.36	1.56	13.53	5.59	22.46	1.58	2.08
Glycolic acid derivative	0.4	1.41	1.14	7.53	-	10.70	1.22	1.35
Acetohydroxamic acid	1.89	0.13	0.66	0.45	5.17	0.42	3.40	8.10
Oxalic acid derivative	0.49	0.41	0.55	-	1.27	-	0.28	0.14
Glyceric acid	0.06	0.23	0.30	1.39	1.09	1.43	-	0.23
Lignoceric acid derivative	-	1.35	-	1.69	-	1.99	-	-
Hexacosanoic acid derivative	-	1.65	-	1.77	-	2.23	-	-
Sum of listed compounds	15.11	39.21	21.58	54.82	41.75	58.05	20.29	39.89

* USE—ultrasound assisted extraction; ** AME—active maceration extraction.

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
