# Peer review of "Determination of Organic Compounds, Fulvic Acid, Humic Acid, and Humin in Peat and Sapropel Alkaline Extracts"

_molecules, 2021, doi:10.3390/molecules26102995_

Round 1

Reviewer 1 Report

The authors present the results of their investigation on quantitation of organic compounds, fulvic acid, humic acid and humin in various peat and sapropel samples from Lithuania. Well the results are interesting and could provide additional information to existing literature. However, there are some technical questions that must be answered. The description of the methods and results is not very clear and impacts the understanding of the manuscript. The presentation of methods and results must be significantly improved before the manuscript can be considered for publication. 

  1. Organic compounds determination by GC-MS. The authors described the derivatization procedure and GC-MS conditions in Section 3 Materials and Methods. However, it is not clear how the organic compounds and organic acids are quantified. There is no indication that standard compounds were used.
  2. In conclusion section, the authors stated "Amount of fulvic acid was determined by using UV spectrophotometer". This is very confusing since the results show that fulvic acids were determined by GC-MS. Please clarify.
  3. Section 3.4 Determination of fulvic acid, humic acid and humin. This section is very confusing. Is fulvic acid determined by the method described in this section the same as presented in Table 1? This section should be rewritten and clearly identify what each step of the procedure does.
  4. The level of decamethylcyclopentasiloxane is presented in Table 1. How is this compound determined? It is not shown in Figure 1. 
  5. Table 1 shows that bis(tert-butyldimethylsilyl)carbonate has the highest level among various organic compounds, and it is labeld as peak 2 in Figure 1. However, the chromatogram does not show peak 2.
  6. The authors used numerous abbreviations. All the abbreviations should be spelt out when used for the first time.

Reviewer 2 Report

The Author should also consider the following article, recently published in Natural Product Research.

Maria Karpińska, Jacek Kapała, Agnieszka Raciborska, Grzegorz Kulesza, Anna Milewska & Stanisław Mnich. Radioactivity of natural medicinal preparations contained extracts from peat mud available in retail trade used externally. Natural Product Research, Volume 31, 2017 - Issue 16. Published online: 30 Nov 2016

Reviewer 3 Report

In order to be published in the Molecules, the manuscript must be  significantly improved, and major revision is required. My recommendations are:

  1. Some typos must be corrected, including the references.
  2. The discussion regarding the chromatographic analysis must be improved; the results are not presented in a clear manner and  certain comparisons with different cited papers are rather confusing. 
  3. Beside chromatogram, please insert a table with all the obtained compounds, retention time and area
  4. Lines 132-134 need citation.
  5. Lines 250-252...Please clarify the quantitative determination of Ca, Fe and Al in the sample. Provide details regarding equipment, method and obtained results; also clarify which is the importance of these determinations for this study.
  6. In the conclusion chapter - lines 261-262- the authors stated that "Amount of fulvic acid was determined by using UV spectrophotometer and humic acid and humin by - 261 termo gravimetrical studies",  and the presented results do not support this statement. In this case also please provide details regarding the used equipment, method and obtained results.
  7. Please improve the conclusion chapter

Reviewer 4 Report

Dear Authors,

Your manuscript is interesting, but to make it clearer to the reader what methods you have used, it would be better to put the Methods and Materials section before Results and Discussion. In my opinion, it will be a little difficult for the reader to compare the results of the two methods before getting acquainted with them and the discussion you have had about this data. 

Round 2

Reviewer 1 Report

The authors have made modifications to the manuscript in response to my review comments. I have two comments on the revised manuscript:

  1. I had a question on the quantitation of the organic acids using GC/MS in the previous round of review; however, the authors did not directly answer the question. Section 3.3 in Section 3 Materials and Methods does not clearly indicate how quantitation was done. My question was whether quantitation was done using an external or internal standard method?Figure 1 provides the peak area of each compound. Was quantitation done based on area percent%? If this is the case, the authors need to discuss whether all the compounds in Figure 1 have similar MS response. This question needs to be answered before the manuscript can be accepted from publication.
  2. Table 1 mentions the derivatives of lactic acid, glycolic acid, oxalic acid, lignoceric acid and hexanosanoic acid. What are these derivatives? There are more peaks in Figure 1 than the compounds in Table 1. How are the peaks in Figure 1 related to the compounds in Table 1?

Reviewer 3 Report

The manuscript is presented in an improved manner Still minor corrections are needed: some English corrections and instead of figure 1 (which has a bad quality) please transform it in a simple chromatogram (with higher resolution) and a table (with RT, areas and compounds names).
